# MEASURING NUMERICAL COMMON SENSE: IS A WORD EMBEDDING APPROACH EFFECTIVE?

## ABSTRACT

Numerical common sense (e.g., "a person with a height of 2m is very tall") is essential when deploying artificial intelligence (AI) systems in society. To predict ranges of small and large values for a given target noun and unit, previous studies have implemented a rule-based method that processed numeric values appearing in a natural language by using template matching. To obtain numerical knowledge, crawled textual data from web pages are frequently used as the input in the above method. Although this is an important task, few studies have addressed the availability of numerical common sense extracted from corresponding textual information. To this end, we first used a crowdsourcing service to obtain sufficient data for a subjective agreement on numerical common sense. Second, to examine whether common sense is attributed to current word embedding, we examined the performance of a regressor trained on the obtained data. In comparison with humans, the performance of an automatic relevance determination regression model was good, particularly when the unit was yen (a maximum correlation coefficient of 0.57). Although all the regression approach with word embedding does not predict values with high correlation coefficients, this word-embedding method could potentially contribute to construct numerical common sense for AI deployment.

## 1 INTRODUCTION

We live in a world where common sense is a basic requirement: if a person's weight is under 50 kg, then they are likely to be very thin; if a person's weight exceeds 80 kg, then they are likely to be hefty. A similar concept can be extended to situations involving money. If a new high-spec laptop is $300, then it is considered to be quite cheap. Therefore, it is essential to account for common sense in future artificial intelligence (AI) systems.

In a society that robots communicate with humans, these robots have to estimate how large the thing is (e.g., If a robot mistakenly estimate the object weight, it may harm humans and/or robots themselves).

There are quantitative units such as physical scales (meter, kilogram, etc.) and money; studies have treated these scales as a unit.

Distributed semantic representations have gained much attention and showed a great performance in varieties of NLP tasks. Mikolov et al. (2013) developed a skip-gram model to obtain semantic representations by using a novel word-embedding approach. Their distributional word embedding exhibited a remarkable performance in analogy tasks (e.g., "*King - Man + Woman = Queen*").

In more recent studies, Gladkova et al. (2016) prepared varieties of analogy pairs named the Bigger Analogy Test Set (BATS) and compared GloVe (Pennington et al. (2014)) and count-based (i.e., PPMI + SVD) model (Church & Hanks (1990)). In our point of view, we assume word themselves have numerical information. That is, similar words share similar units and thus concrete words contains numerical information. The primary motivating assumption is that, similar to analogy, contextual flow in word sequences would have specific repetitive representations, and thus we assume that this specific word representation (e.g., co-occuring unit word such as *gram*) would include the numerical knowledge for a given target object.

Here, we attempt to analyze the relationship between human-generated and extracted common sense by using word embedding. We assume that common sense is attributed to distributional word embeddings such that a trained regressor can predict numerical values.

The uniqueness in this field lies in treating abstract concepts as common sense. Other than the size of the object, we focus on nonphysical scales, such as temperature and weight, as well as subjective scales, such as money. Moreover, a challenging problem when estimating numerical common sense is that there are few datasets with labeled numerical values. This issue is tackled in our study through a word embedding approach. The contributions we make are two-fold:

- We used a crowdsourcing service to obtain a sufficient number of subjective agreements on numerical common-sense data.

- We examined the predictabilities of regressor trained using word embedding and crowd-sourced data. We also examined whether a concatenated semantic vector can effectively predict numerical values.

The remainder of the paper is organized as follows: After introducing related works in Section 2, we describe the collected datasets in Section 3. The results of the experiments conducted using the obtained data are then presented in Section 4. Subsequently, in Section 5, the obtained results are discussed and interpreted. The study is summarized and future scope is described in Section 6.

## 2 RELATED WORK

### 2.1 APPLICATIONS THAT TREAT NUMERICAL VALUES

Narisawa et al. (2013) proposed methods that use a large amount of text, related to the abovementioned units, to determine common sense text patterns. Their approach estimates the range of the units co-occurring within a context and examines whether a given value is large, small, or normal based on its distribution. Their approach utilizes textual patterns with which speakers explicitly express their judgment regarding the value of a numerical expression. Similarly, Hayakawa & Hagiwara (2014) used snippets from a web search engine to extract more scalable but a lesser amount of text data, related to the units, through the design of heuristic functions that enable the system to handle input frequencies, which are often very noisy. These two approaches require careful intuitions when determining the thresholds of a proposed method or building an effective function for each numerical value. Qin et al. (2018) further developed a numerical variable extraction mechanism. They modeled the joint probability of data fields, texts, phrasal spans, and latent annotations with an adapted semi-hidden Markov model and imposed a soft statistical constraint to further improve the performance of the model.

Another approach focuses on using different modalities. A 165 cm human cannot be taller than a 1 km long bridge. This concept has a wider range of applications. Bagherinezhad et al. (2016) considered object sizes using a computer vision technique.

In the context of only text, particularly in relation to numerical common-sense extraction, most methods rely on large amount of data with carefully designed heuristics. As such, it would be beneficial if it was possible to extract common sense only using certain human evaluations.

Forbes et al. Forbes & Choi (2017) present an approach to infer relative physical knowledge of actions and objects along five dimensions (e.g., size, weight, and strength) from unstructured natural language text. They reported that it is possible to extract knowledge of actions and objects from language and that joint inference over different types of knowledge improves performance. Yan et al. Yang et al. (2018) proposed and assessed methods for extracting one type of commonsense knowledge, object-property comparisons, from pre-trained embeddings. They also showed that an active learning approach that synthesizes common-sense queries can boost accuracy. Elazar et al. Elazar et al. (2019) propose an unsupervised method for collecting quantitative information from large amounts of web data, and use it to create a new, large resource consisting of distributions over physical quantities associated with objects, adjectives, and verbs.

| NCS-50x1 | object | unit | NCS-60x3 | object | unit |
|---|---|---|---|---|---|
| | family member | number | | laptop | yen |
| | milk | ml | | apple | centimeter |
| | winter climate | degree Celsius | | coffee cup | meter |
| | university entrance rate | percentage | | piano | kilogram |
| | annual income | yen | | private jet plane | gram |
| | | | | wine | |

Table 1: Examples of objects and units for each dataset. Note that the object and unit need not correspond.

## 2.2 STUDIES ON DISTRIBUTED SEMANTIC REPRESENTATION

The approach proposed herein is closely related to existing semantic vector models (i.e., word embedding). More recent studies used global contexts Pennington et al. (2014) or incorporated other external data, such as brain activity from fMRI data Fyshe et al. (2014), to improve performance. fastText Bojanowski et al. (2017) improves performance by considering subwords in the model. Jeawak et al. (2019) imcoporated geographical information into semantic representations to deal with locations for flicker images.

## 2.3 THEORETICAL BACKGROUND ON NUMERICAL WORD EMBEDDING

We describe why numerical values are related to word embedding. In the context of distributed word embeddings, "count-base" and "prediction-based" method has something in common. As a traditional count-based method, before conducting SVD, matrices of Positive Pointwise Mutual Information (PPMI) is used to obtain word embeddings (Church & Hanks (1990)). More concretely,

$$\text{PPMI} = max(0, \text{PMI}(x, y)) = \max\left(0, \log_2 \frac{P(x, y)}{P(x), P(y)}\right) \tag{1}$$

where $P(x)$ and $P(y)$ are frequencies of each word and $P(x, y)$ indicates co-occurrence of the two words. As such, a word that co-occur with words in a group and another word that co-occur with words in the smilar group will have similar embedding. Numerical representation will on this point will be obtained in this case.

Similarly, for current prediction-based word embeddings, a loss fucntion of continuous bag-of-words (CBOW) is represented as:

$$L_{\text{CBOW}} = -\frac{1}{T} \sum_{t=1}^{T} \log P(w_t | w_{t-1}, w_{t+1}) \tag{2}$$

where $w_{t-1}, w_{t+1}$ are to predict $w_t$, resulting in surrounding words will have similar embedding. In addition, loss function of Skip-gram is calculated as:

$$L_{\text{Skip-gram}} = -\frac{1}{T} \sum_{t=1}^{T} \left( \log P(w_{t-1} | w_t) + \log P(w_{t+1} | w_t) \right) \tag{3}$$

where $w_t$ is to predict $w_{t-1}, w_{t+1}$. Because Skip-gram has to handle one word to predict multiple words, we assume Skip-gram will obtain more information about numerical values.

## 3 DATA COLLECTION

We used the Japanese crowdsourcing service Lancers to collect data on numerical common sense[1]. Table 1 lists examples of objects and units for each dataset.

We asked participants to annotate these objects and units to indicate "from which value you think it's large" and "from which value you think it's small." Note that for temperature, this "large quantity"

---

[1]We plan to make the dataset available to the public after publication

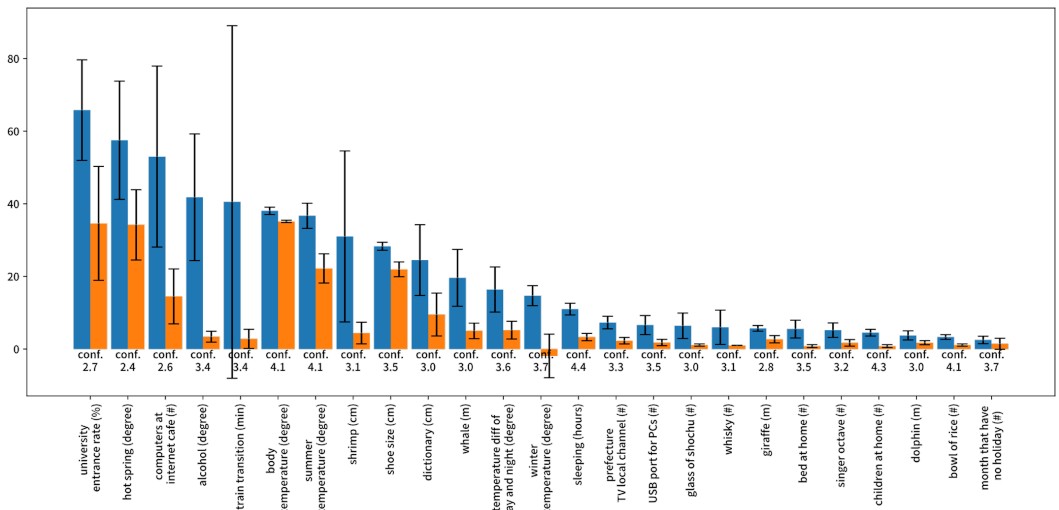

Figure 1: Some examples from NCS-50x1. The histogram indicates the items' averages and the bars represent the standard deviations.

will be "high temperature." For Numerical Common Sense including 50 items with one pair (NCS-50x1), we asked 10 participants to create 50 pairs of samples with different units. For Numerical Common Sense including 60 items with three pairs (NCS-60x3), we asked 10 different participants to construct 60 pairs of samples with three different units. In a total, we collected 50 + 60 x 3 = 230 pairs of units and values. We also collected participants' confidence (5-rated evaluation: 1 is least confident and 5 is most confident) for their answers.

### 3.1 CONSTRUCTION OF NCS-50x1

In NCS-50x1, we selected terms that represent an "object" and "unit" based on the results obtained by Hayakawa & Hagiwara (2014) and included all their 26 examples. Moreover, we added other 24 object and unit pairs. This dataset was collected on February 28, 2019.

Figure 1 illustrates some examples that were collected. The ranges of "small" and "large" differ depending on the term. For example, large in train transition is around 10 times more (with a large variance) than small, while both large and small in body temperature have smaller variances. Moreover, there are terms whose confidence from participants is more than others. Compared with guessing large/small of temperatures in hot spring, participants are confident when predicting large/small of sleeping hours.

### 3.2 CONSTRUCTION OF NCS-60x3

In NCS-60x3, for a more exhaustive study on the variation concept, we limit objects and units using WordNet Miller et al. (1990) and brain-based knowledge representation Naselaris et al. (2009) as a reference. In the supplementary material provided by Naselaris et al. (2009), a handful of word concepts to check the brain decode and encode predictabilities were shown; we assumed these concepts could be applied to crowdsourcing on a reasonable scale. We use 60 objects considering the above concepts and the obtained crowdsourced data. Particularly, animate, artificial, and food objects were taken into account. This dataset was collected on March 3, 2019.

We depict the histogram of 18 example items in Figure 2 and 3.

As the figure shown, the results from participants are very diverged. To make a fair comparison, we use the same unit through size, and weight (i.e., *m* is converted to cm by computing "x 100").

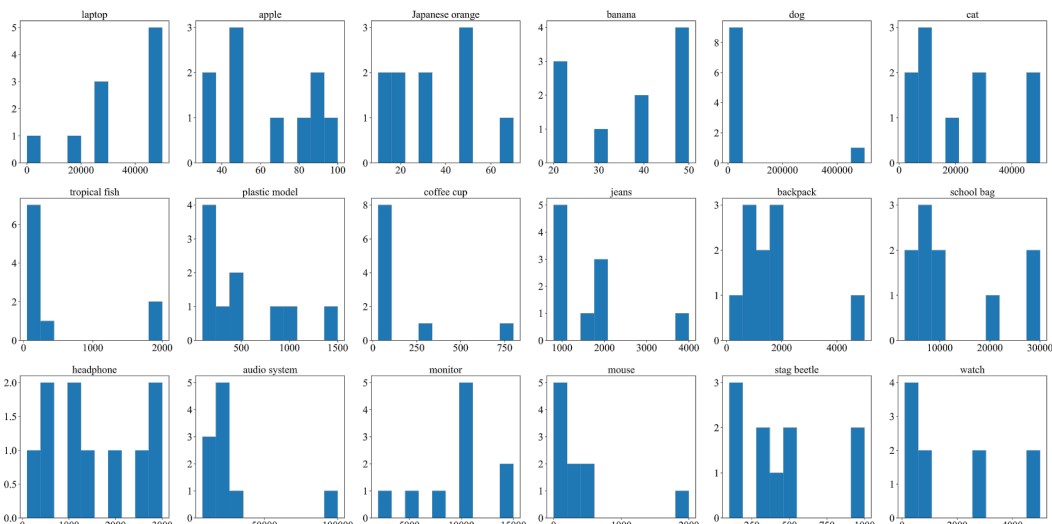

Figure 2: Histogram of *money-small* on 18 out of 60 items. The X-axis is *Yen* and the Y-axis represents number of participants.

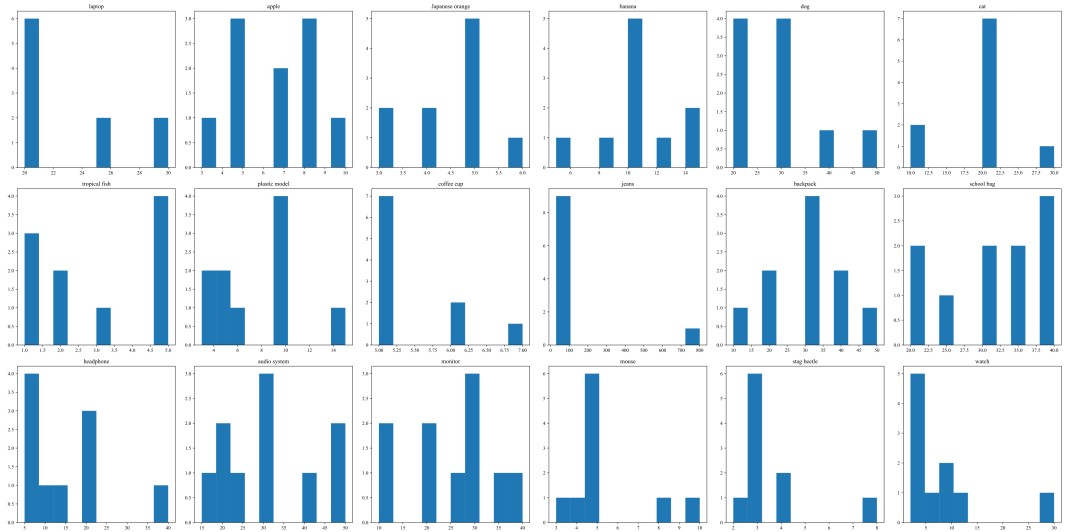

Figure 3: Histogram of *size-small* on 18 out of 60 items. The X-axis is *centimeter* and the Y-axis represents number of participants.

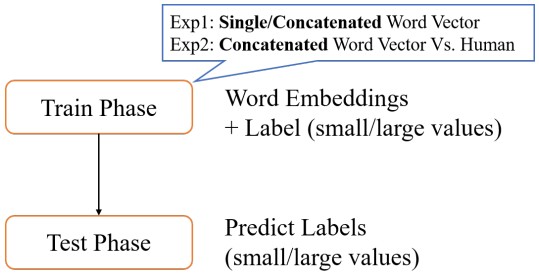

Figure 4: Diagram of experimental procedure

# 4  EXPERIMENT FOR NUMERICAL VALUE PREDICTION

We analyze regressors trained on the word embeddings along with human evaluated data to examine whether common sense is attributed to current word embedding. Figure 4 shows a diagram of this experiment. After checking performance on various units with/without word embedding concatenation on different regressor in Experiment 1, we compare the regression performance on each word embedding type trained on different corpus. Then we also compare that performance with humans in Experiment 2. We intend to 1. investigate overall tendencies using NCS-50x1 in the first experiment because we include terms (objects and units) from one of the pioneer works of numerical common sense (Hayakawa & Hagiwara (2014)), and 2. obtain more detailed points including results on selected units and types of word embeddings comparing with human participants in the second experiment.

Section 3.1 provides more details on the experimental settings and Section 3.2 describes experiments 1 and 2, where we use NCS-50x1 and NCS-60x3, respectively.

## 4.1  EXPERIMENTAL SETTINGS

We describe the details about word embeddings and machine learning algorithms that we use. We followed the procedure described below to prepare for the experiment.

- We use word embeddings trained on Japanese Wikipedia, which was downloaded in Feb 2015. To simplify the problem, when there were multiple words in an object, feature vectors were averaged. Note that when tokenizing, we used MeCab (Kudo et al. (2004)). We then obtained word embeddings based on a Continuous Bag-of-Words (CBOW) model with 200 dimensions. Henceforth, we refer to this as "w2v-wiki."

- We use fastText word embeddings Bojanowski et al. (2017) trained on Japanese Wikipedia that are available at Grave et al. (2018). The model was trained using CBOW with position-weights, in dimension 300, and character n-grams of length 5, a window of size 5 and 10 negatives. Henceforth, we refer to this setting as "fT-wiki."

- We use word embeddings trained on crawled Japanese web texts (Yata), Nihongo Web Corpus (NWC). The texts are composed of approximately 100 million html documents (3.25TB in total), which resulted in 396 GB of plain texts. Note that when tokenizing, we used MeCab with the NEologd dictionary. The model was trained using CBOW, in dimension 500, with a window of size 15. We refer to this setting as "w2v-nwc."

- For experiment 1, we prepared different types of word vectors; *object only*, *unit only*, and *concatenated feature*, respectively. For *object only* (e.g. "suimin": *sleeping*) and *unit only* (e.g., "jikan": *hours*), word embeddings are directly extracted from "w2v-wiki" and regressors are trained and tested. For *concatenated features*, those two word vectors (namely *object* and *unit*) were concatenated, which resulted in 400 dimensions for the concatenated features. For Experiment 2 where we use NCS-60x3 dataset, we also use "fT-wiki" whose concatenated embedding dimension is 600, and "w2v-nwc" whose one is 1000.

- We employed a linear regressor with the least square (LS) error penalty and a linear regressor with an automatic relevance determination (ARD) prior. We also use random forest

| object only | Mean Abs. Err. | Corr. Coef. |
|---|---|---|
| ARD | $1.20 \times 10^7$ | 0.080 |
| LS | $1.15 \times 10^7$ | 0.011 |
| RF | $1.04 \times 10^7$ | 0.002 |
| **unit only** | | |
| ARD | $1.20 \times 10^7$ | N/A |
| LS | $1.10 \times 10^1$ | N/A |
| RF | $9.91 \times 10^6$ | N/A |
| **Concat. feat.** | | |
| ARD | $1.20 \times 10^7$ | 0.254 |
| LS | $1.30 \times 10^7$ | -0.035 |
| RF | $9.08 \times 10^5$ | 0.181 |

Table 2: Results obtained from linear and random forest regressors. Note that Corr. Coef. indicates Pearson's correlation coefficient and MAE is mean absolute error

(RF) regressor for comparison. In the RF, the number of decision trees was set to 10; in ARD, the parameters were set to alpha1=1e-06, alpha2=1e-06, lambda1=1e-06, and lambda2=1e-06. The LR input will be normalized before regression by subtracting the mean and dividing by the l2-norm. After average numerical values were obtained from 10 subjects, we conducted a 10-fold cross validation. To examine performances of those regressors, we use the mean absolute error (MAE) and the Pearson's correlation coefficient.

- To compare the predictability between CBOW and Skip-gram, and increament of word embedding dimensions, we use Japanese Wikipedia and change the dimensions from 100 to 500 in experiment2.

- We used gensim Rehurek & Sojka (2010) for training word embeddings and feature extraction, and scikit-learn Pedregosa et al. (2011) for executing machine learning algorithms.

  We use w2v-wiki for Experiment 1 and all three word embeddings for Experiment 2.

## 4.2 EXPERIMENTAL RESULTS

### 4.2.1 EXPERIMENT 1 ON NCS-50x1

The regressor performance results are shown in Table 2. Essentially, MAE showed high prediction errors and the Pearson's correlation coefficient was quite low. The *Concat feat.* in ARD appeared to have a higher correlation than other non-human methods. Therefore, in the next experiment we employ word embedding concatenation.

### 4.2.2 EXPERIMENT 2 ON NCS-60x3

Table 3 list the results for 60 objects with three different fixed units obtained from linear regressors.

Here, we fixed a unit and examined the predictabilities of word vectors trained on human evaluation. In Experiment 2, we also compared the results of the trained regressor with those of human evaluation.

Humans scores and predictors are calculated in the same manner. Concretely, in a 10-fold cross validation manner, correlation between evaluation scores from each subject and averaged other 9 subjects were calculated. Humans performed better than both of regressors, yet ARD had a correlation coefficient of 0.57, which appeared to be comparable to that of humans when the unit is *Yen* and *gram*.

In most cases, the regressor had better prediction in MAE when the unit was *centimeter*. This may be because there is less variation in values. In contrast, the correlation coefficient of both regressors showed better performance for the unit of *gram*.

In addition, we examined CBOW and Skip-gram with an increase in dimensions. When we tested the data on Japanese Wikipedia, we observed that the accuracy changed with an increase in the dimensions For ADR, the average correlation coefficient across units [*Yen*, *g*, *cm*] was 0.213 with 200-d, 0.186 with 300-d, 0.221 with 400-d, and 0.221 with 500-d. We observed better correlation

| | MAE | | | Corr. Coef. | | |
|---|---|---|---|---|---|---|
| *unit: YEN* | w2v-wiki | fT-wiki | w2v-nwc | w2v-wiki | fT-wiki | w2v-nwc |
| large on ARD | **9.07 x $10^7$** | **9.07 x $10^7$** | **9.07 x $10^7$** | **0.360** | -0.237 | **0.571** |
| small on ARD | **7.56 x $10^6$** | **7.56 x $10^6$** | **7.56 x $10^6$** | **0.403** | 0.013 | **0.418** |
| large on LS | 2.89 x $10^8$ | 3.80 x $10^8$ | 1.85 x $10^8$ | 0.216 | -0.033 | 0.444 |
| small on LS | 2.45 x $10^7$ | 3.39 x $10^7$ | 2.45 x $10^7$ | 0.212 | -0.118 | 0.268 |
| large on RF | 1.38 x $10^8$ | 7.56 x $10^7$ | 1.49 x $10^8$ | 0.127 | **0.217** | 0.500 |
| small on RF | 1.14 x $10^7$ | 4.40 x $10^6$ | 1.29 x $10^7$ | 0.119 | **0.262** | -0.012 |
| large on human | | 6.08 x $10^7$ | | | 0.753 | |
| small on human | | 4.73 x $10^6$ | | | 0.894 | |
| *unit: cm* | | | | | | |
| large on ARD | 712 | 911 | 695 | 0.120 | 0.0070 | **0.310** |
| small on ARD | 51.4 | 189 | **142** | 0.205 | -0.018 | **0.487** |
| large on LS | 383 | 793 | 479 | **0.502** | 0.111 | 0.249 |
| small on LS | 31.9 | 284 | 142 | 0.270 | -0.014 | 0.460 |
| large on RF | **339** | **189** | 117 | 0.236 | **0.259** | 0.067 |
| small on RF | **25.1** | **105** | 102 | **0.148** | **0.162** | 0.234 |
| large on human | | 312 | | | 0.828 | |
| small on human | | 51.9 | | | 0.905 | |
| *unit: g* | | | | | | |
| large on ARD | **3.99 x $10^4$** | 3.98 x $10^7$ | 3.98 x $10^7$ | **0.508** | -0.371 | **0.359** |
| small on ARD | **9.21 x $10^3$** | 9.25 x $10^6$ | **9.25 x $10^6$** | **0.339** | -0.265 | **0.358** |
| large on LS | 1.27 x $10^5$ | 2.00 x $10^8$ | 8.33 x $10^7$ | 0.180 | 0.182 | 0.118 |
| small on LS | 2.08 x $10^4$ | 2.56 x $10^7$ | 1.47 x $10^7$ | -0.075 | -0.266 | 0.060 |
| large on RF | 5.52 x $10^5$ | **2.60 x $10^7$** | **1.46 x $10^6$** | -0.047 | -0.041 | 0.041 |
| small on RF | 1.20 x $10^4$ | **8.61 x $10^6$** | 1.29 x $10^7$ | 0.292 | 0.092 | 0.288 |
| large on human | | 3.84 x $10^7$ | | | 0.746 | |
| small on human | | 8.41 x $10^6$ | | | 0.774 | |

Table 3: Mean absolute error results for 60 objects with 3 different fixed units obtained from linear and random forest regressors with unit on the left (*Yen*, *cm*, and *g*). Pearson's correlation coefficient results on the right. Bold digits in each row indicate higher performance.

when using Skip-gram for ARD regression. Average correlation coefficient across units was 0.343 with 200-d, 0.378 with 300-d, 0.325 with 400-d, and 0.372 with 500-d.

## 5 DISCUSSION

We computed the predictabilities of numerical values solely from word embeddings to examine whether they have such information. This idea is based on whether corpus has some contextual information of numerical values. Because word embedding models treat context windows, when predictabilities are obtained, some kind of numerical knowledge is embedded as co-occurrence. Word embedding is treating co-occurrence and numerical values may be included. In context, objects that are occur is similar and then, text embedding will be possible because the numerical tendencies will be similar. One can imagine that w2v-nwc embedding outperformed in correlation coefficient because of not only its size and diversity, but also contextual window size.

Overall, predicting large was quite diverged on both on regressors and humans.

As indicated by Figure 1, numerical values from annotators vary for specific words. As depicted in Figure 2, some metrics especially size had a large variance. For example, the size of some items (e.g., "plastic model" and "audio system") differed more than 1000 times depending on persons when treating *large*. Basically, it was also difficult to predict money when it comes to MAE. A few unknown but expensive objects such as "private jets" considerably influenced on the result. While our method on MAE did not work well, correlations between predicted numerical values and human evaluations were observed. Prediction by regressor is quite comparable to human in terms of correlation coefficient. As we anticipated, Skip-gram showed a better performance.

## 6 CONCLUSION

We are living in a world where we need common sense; thus quantifying numerical common sense is essential to deploy AI systems in society. We collected and analyzed numerical common-sense values through a crowdsourcing web service. Our contributions are two-fold.

First, we collected data on a subjective yet not fully explored area. We then compared current approaches from a quantitative perspective. To this end, we first used a crowdsourcing service to obtain sufficient data for a subjective agreement on numerical common sense. We constructed two datasets, namely, NCS-50x1 and NCS-60x3.

Second, we compared the averaged numerical values from participants on the crowdsourcing platform with that of the regressor using word vectors. To the best of our knowledge, this is the first approach that tackles this point. In comparison with humans, the performance of an automatic relevance determination regression model was good, particularly when the unit was *yen* (up to a correlation coefficient of 0.57). In addition, predictability increased when we employed the Skip-gram. Even though we confirmed positive correlation coefficients, the model was not comparable with the data collected using the crowdsourcing service when it was trained on different units (e.g., when trained on yen and tested on *gram*).

Although our novel regression approach with word embedding does not always predict values with high correlation coefficients, it exhibited potential to quantify numerical common sense.

In addition, obtaining real numerical distributions would be very helpful in advancing this area. We have a number of large copora with numerical numbers. For example, Wikidata, which was converted from Wikipedia to be precessed by computers can be useful as an extention. We believe those datasets can be utilized to expand the size of the numerical common sense dataset. Moreover, it would be more effective to employ more powerful newural regressors.

Simultaneously, we plan to pursue the sophistication of semantic representations of the model.

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

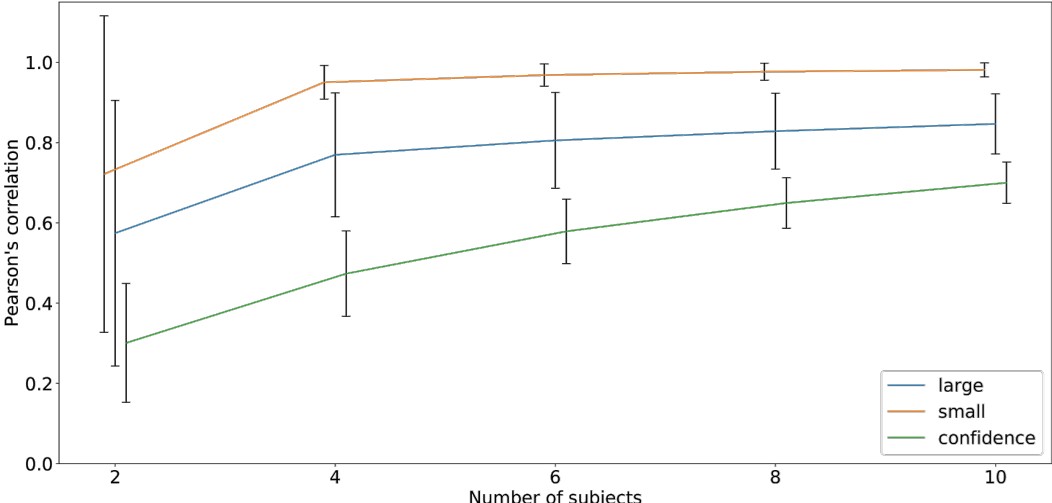

Figure 5: Inter-rater agreements of NCS-50x1

# A  APPENDIX

## A.1  DETAILS OF DATASETS

### A.1.1  NCS-50x1

To check the agreement on each participant, we conducted a correlation analysis as shown in Figure 5.

To calculate the inter-rater agreements, we aggregated the scores from participants, and we computed correlation between groups of subjects. In the figure, the X-axis indicates the number of subjects and the Y-axis is the calculated correlation. The bars indicate the standard deviation of the correlation. As the number of the participants increases, reasonably high correlation is observed. Compared with *large* and *confidence*, *small* had higher correlation.

### A.1.2  NCS-60x3

The results of a correlation analysis are shown in Figure 6 when the number of participants increases.

In the figure, the X-axis indicates the number of subjects and the Y-axis is the calculated correlation. The bars indicate standard deviation of the correlation. Except for *size large*, as the number of participants increases, higher correlations were obtained. Because a few items (e.g., "plastic model" and "audio system") that are not common for participants were included, it seems that estimating sizes, especially *large* was quite diverged. Compared with other modalities, prices shared similar tendencies, which resulted in a higher correlation.

Herein, we list results of histograms. Note that these represent values in NCS-60x3 dataset and showing "price-large", "price-small", "size-large", "size-small", "weight-large", "weight-small." and confidence with 5-point scales. Units (which corresponds to the X-axis) are "Yen", "Yen", "cm", "cm", "g", "g", and "1-5" respectively for each page in all figures. Note that the number of the participants is 10 so that the maximum number of the Y-axis will be 10 at the maximum. Figure 7, 8, 9 10, 10, 11, 12, and 13 show histograms of price-*large*, price-*small*, size-*large*, size-*small*, weight-*large*, weight-*small*, and *confidence*, respectively.

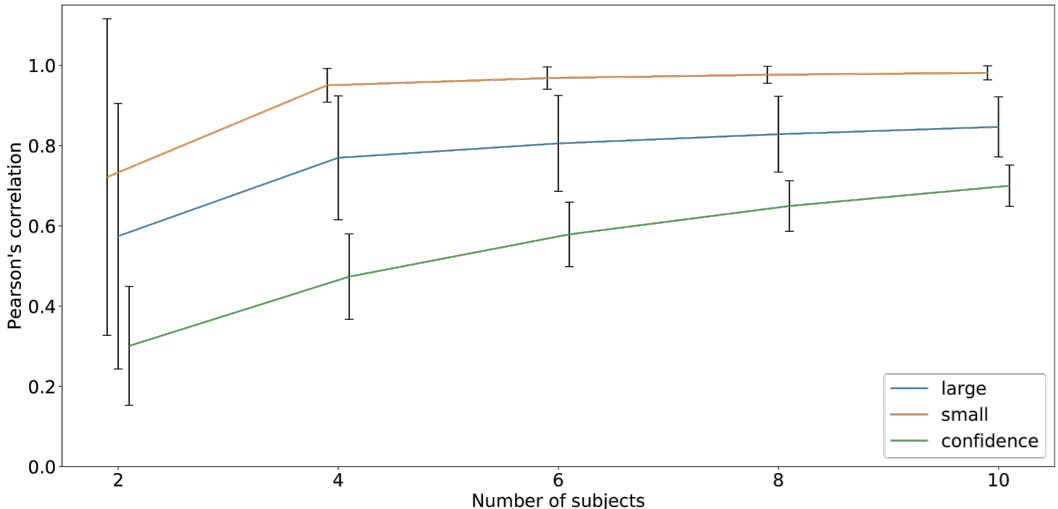

Figure 6: Inter-rater agreements of NCS-60x3

## A.2 SCATTER PLOT OF EXPERIMENT

We show a scatter plot of test human and prediction in Figure14. Note that we conducted the 10 fold-cross validation for prediction.

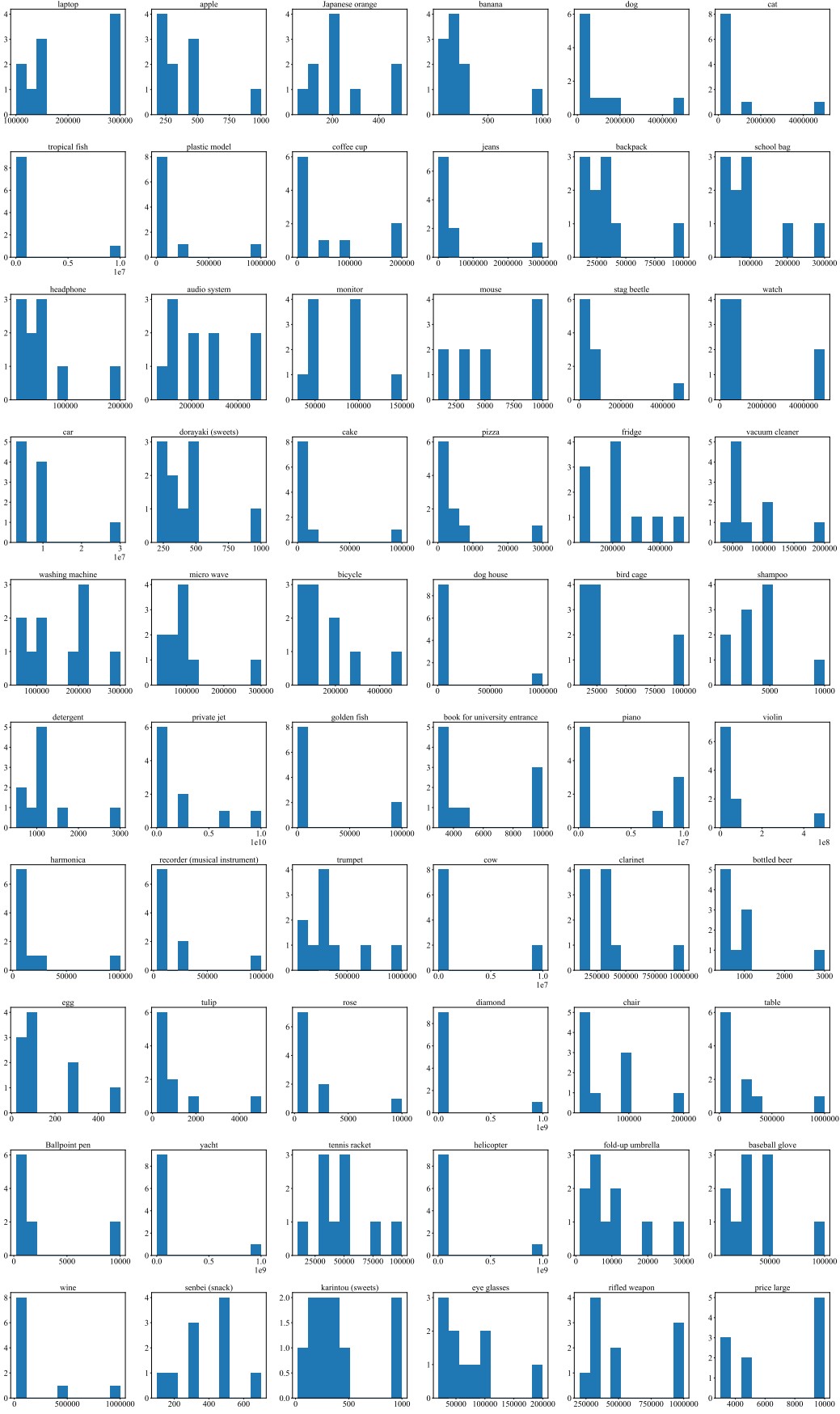

Figure 7: Histograms of price-*large*

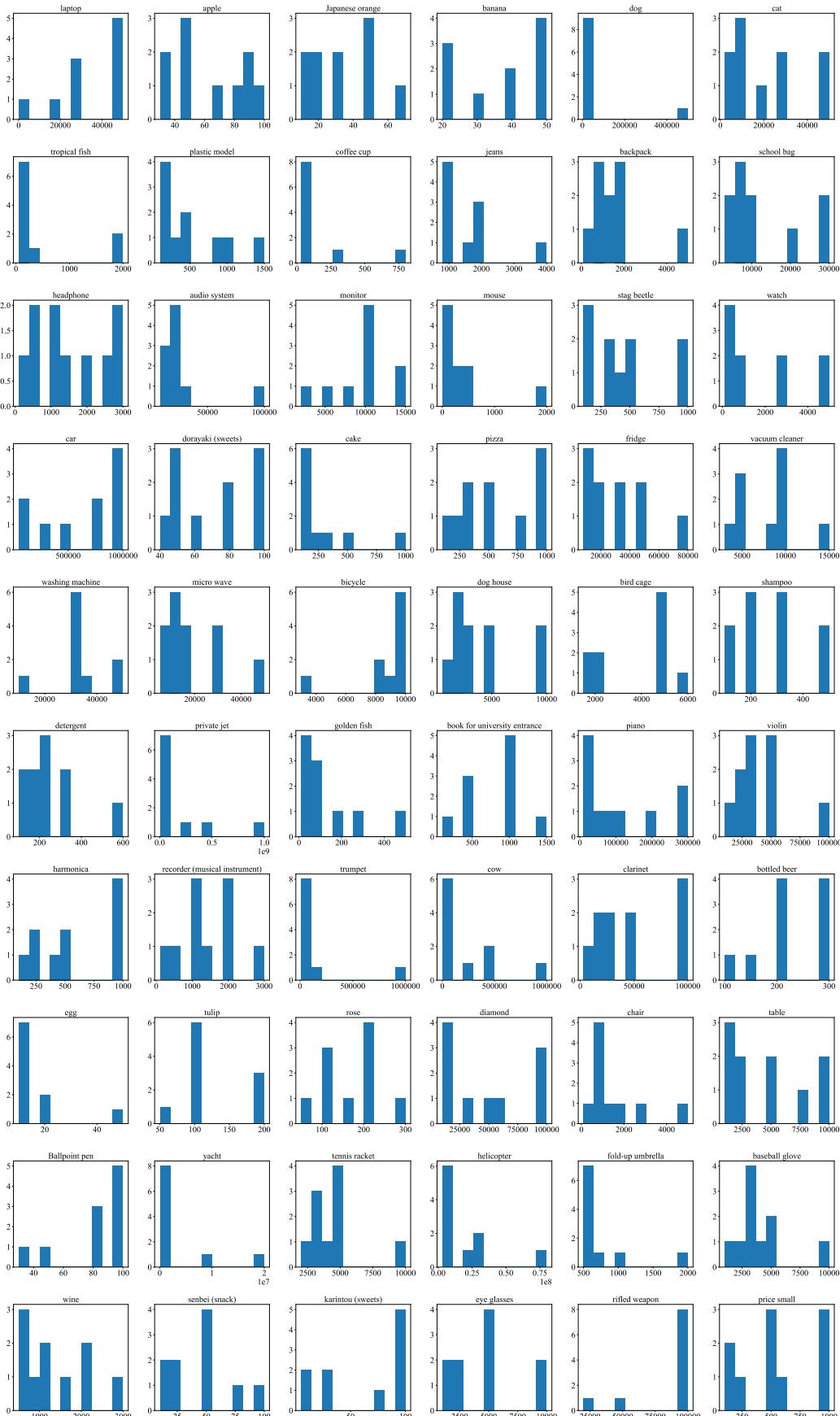

Figure 8: Histograms of price-*small*

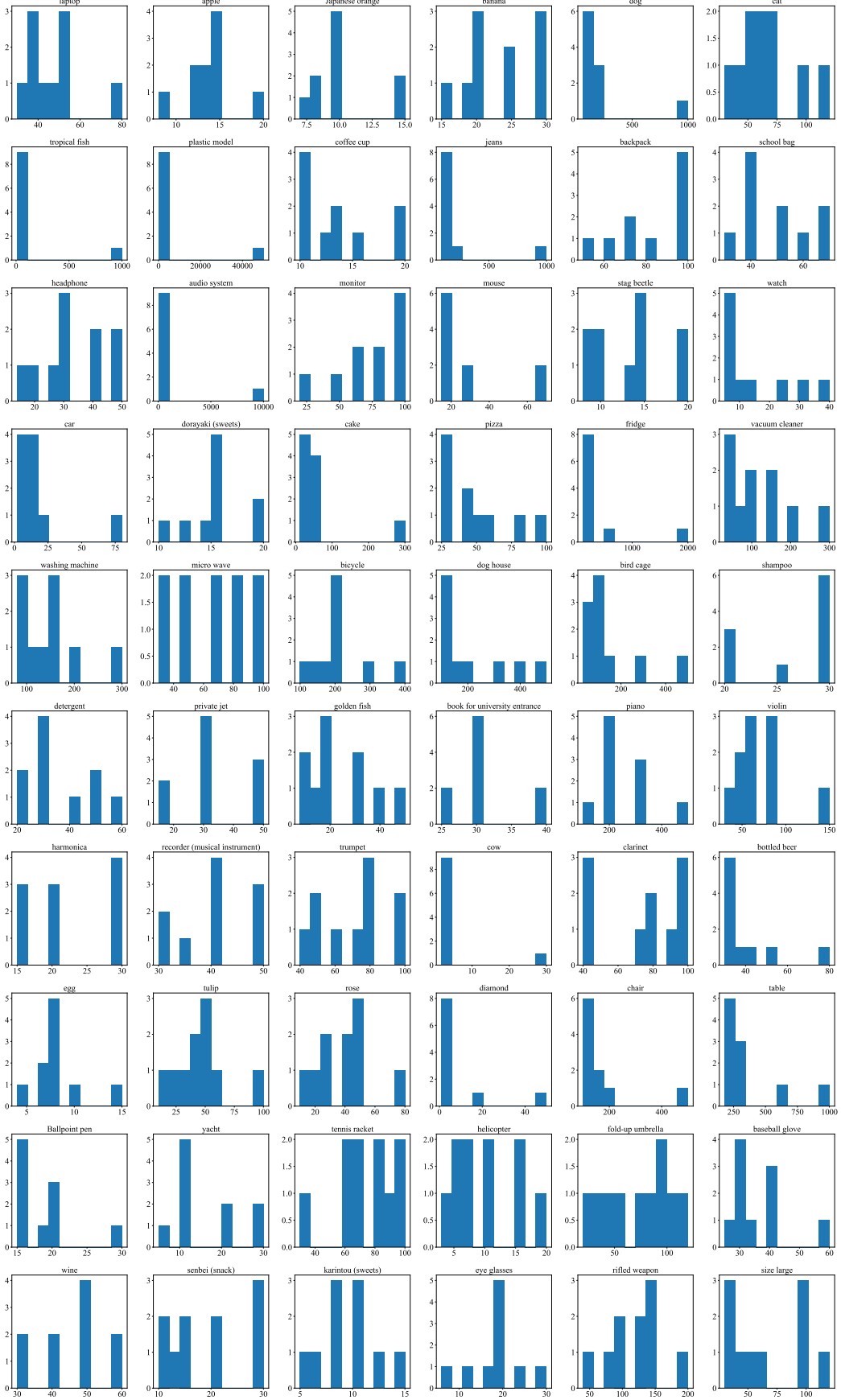

Figure 9: Histograms of size-*large*

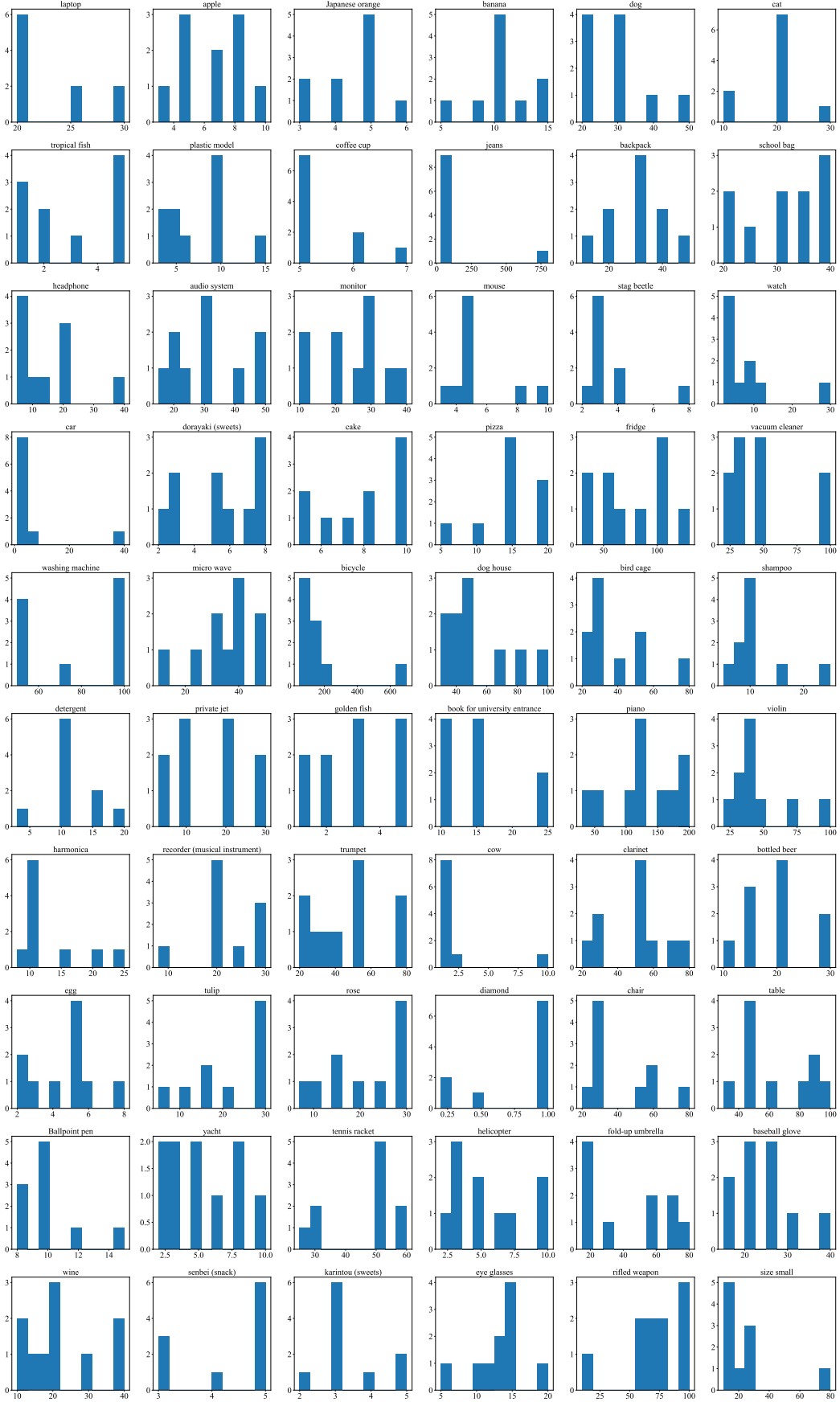

Figure 10: Histograms of size-*small*

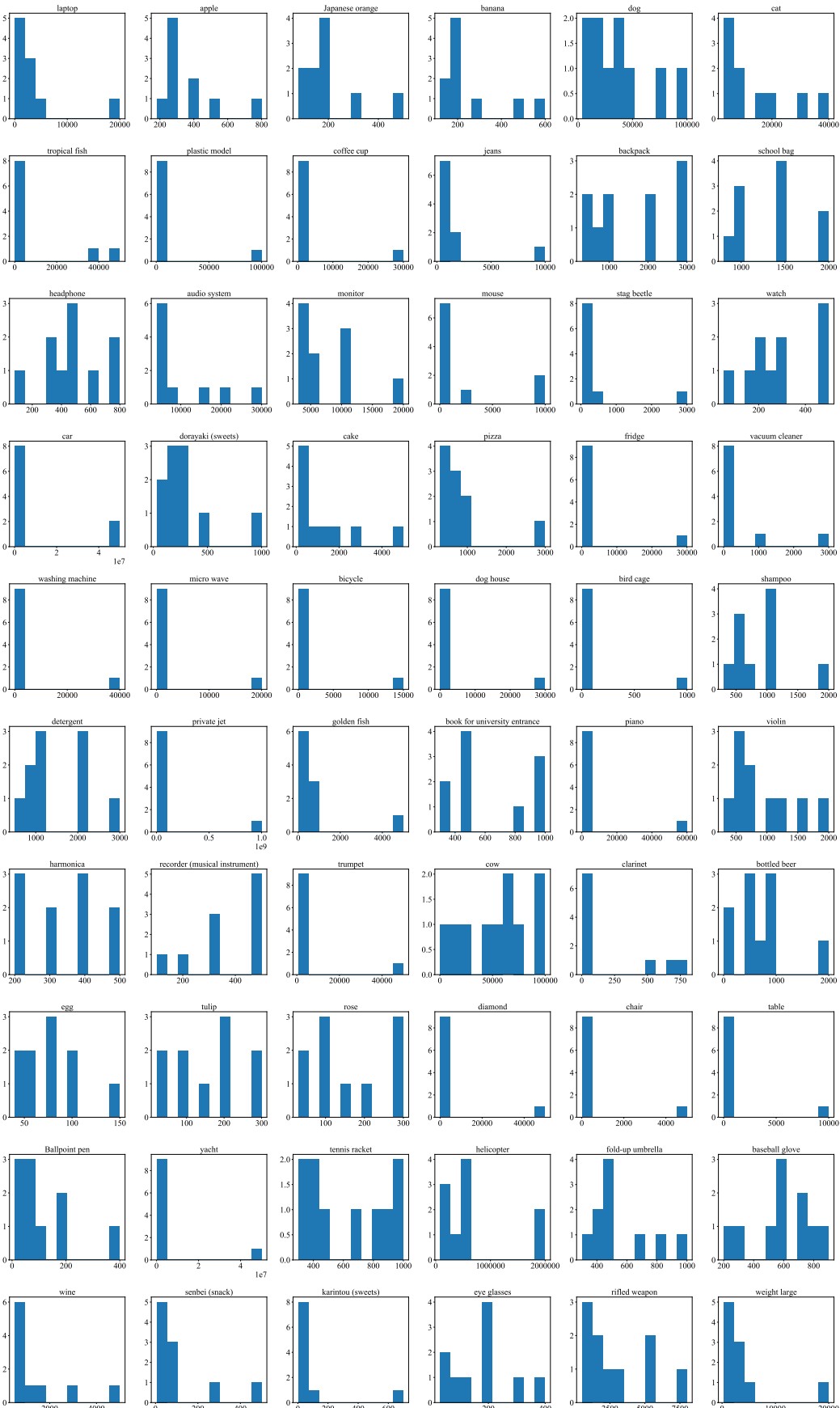

Figure 11: Histograms of weight-*large*

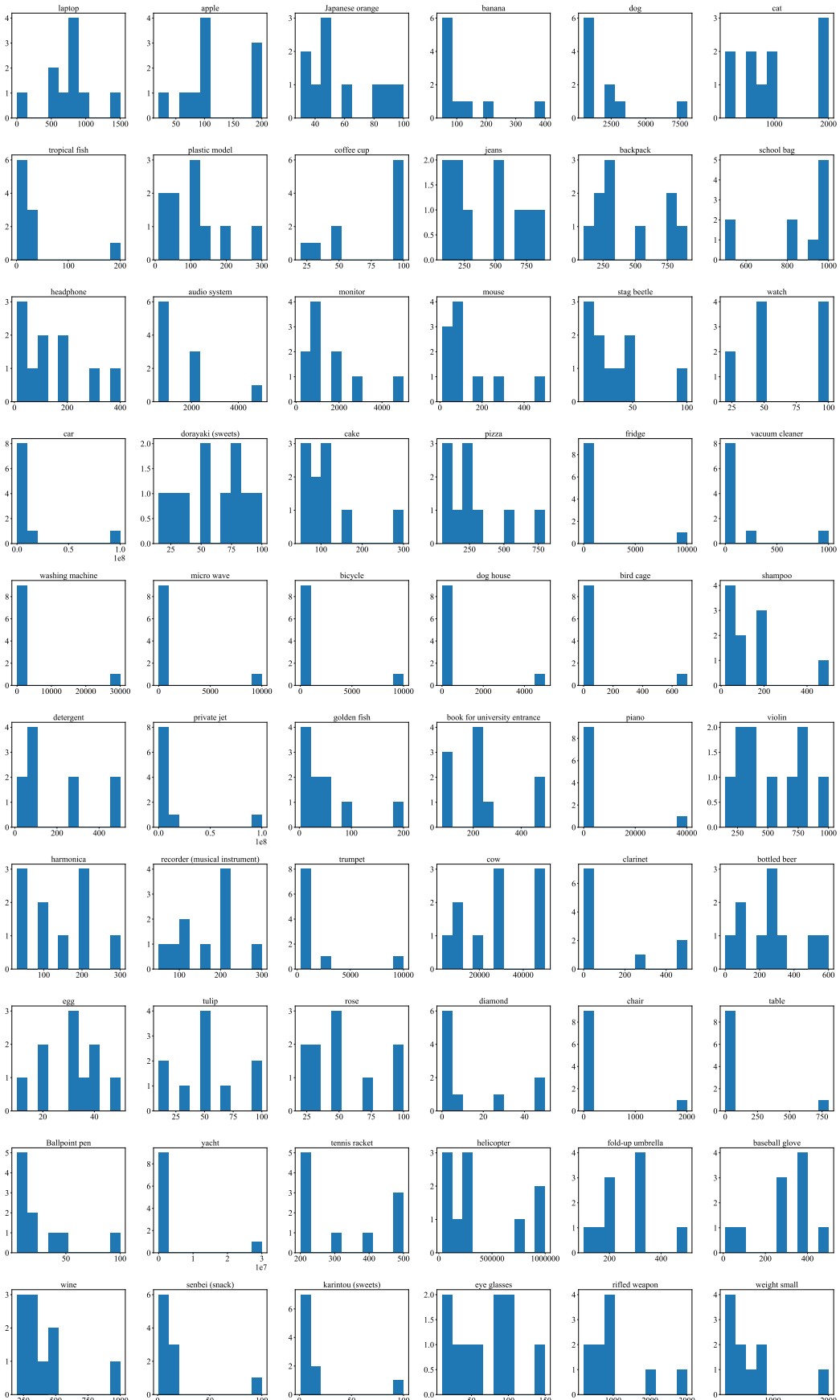

Figure 12: Histograms of weight-*small*

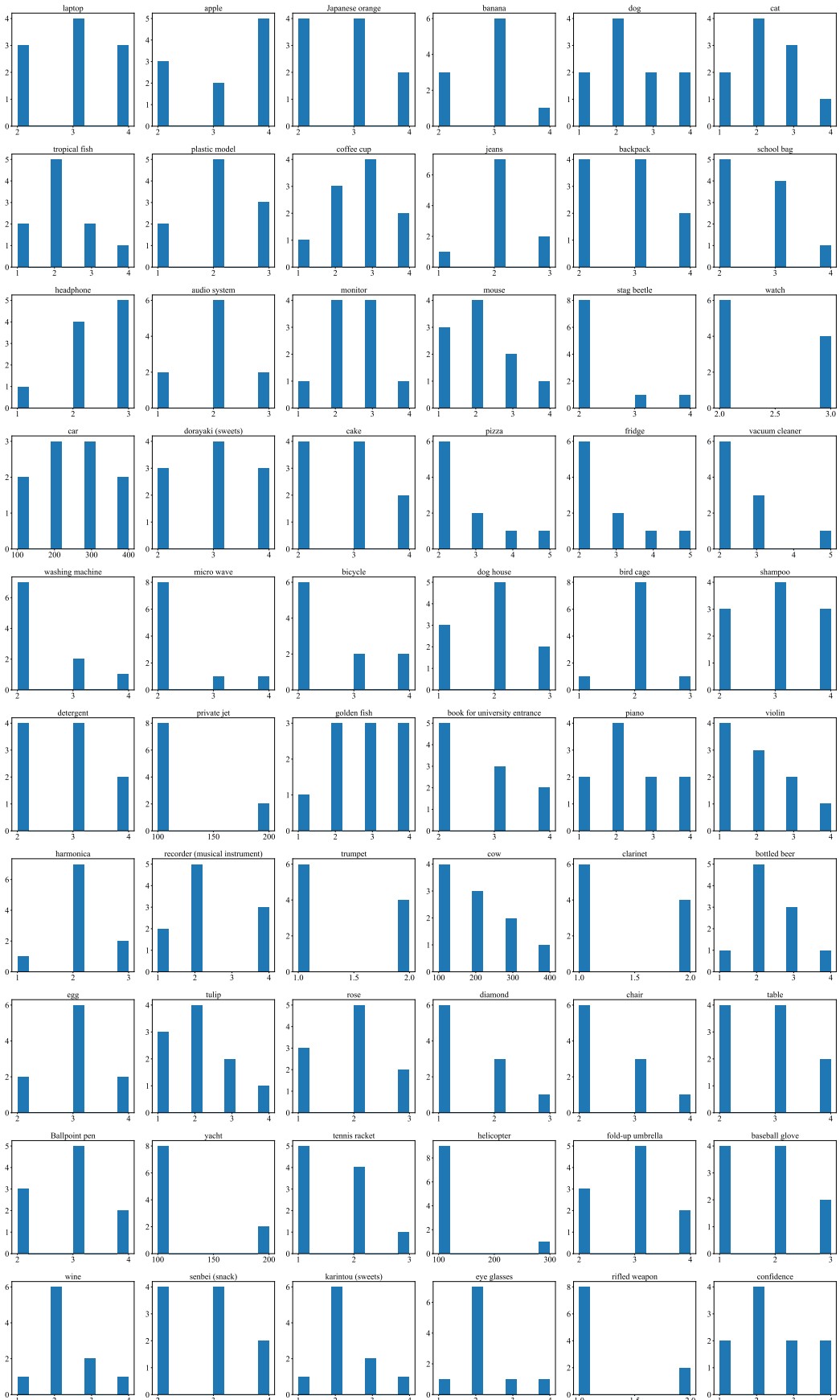

Figure 13: Histograms of *confidence*

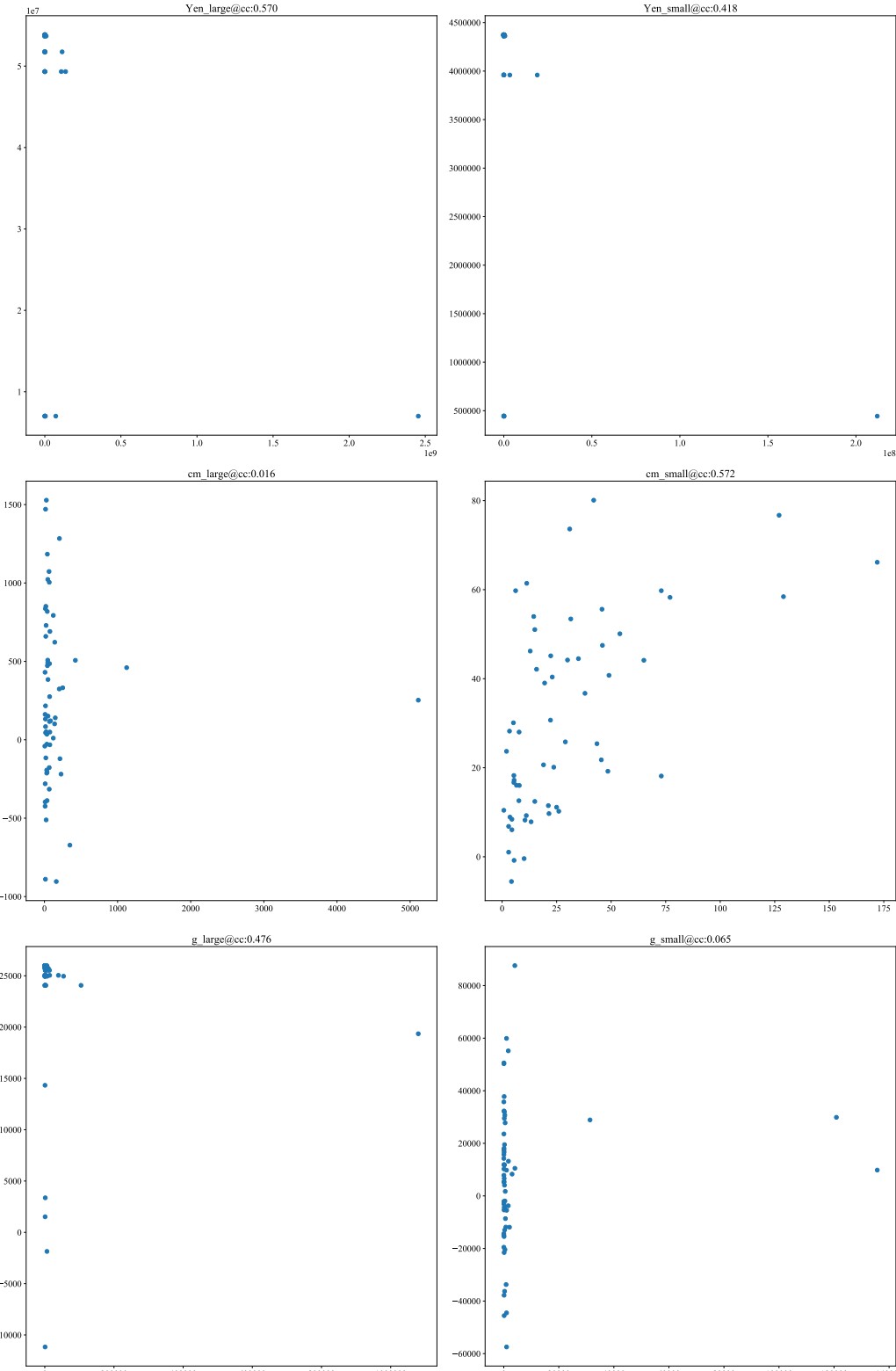

Figure 14: X-axis is test (human) and y-axis is prediction by ARD regressors

