# OpenReview forum: "Measuring Numerical Common Sense: Is A Word Embedding Approach Effective?"
_ICLR.cc/2020/Conference — Reject_

### Official Review · AnonReviewer1 · 2019-10-22
**Official Blind Review #1**

**Rating:** 1

**Review:**

I have read the author response.  Thank you for responding to my questions.

This paper aims to predict typical “common sense” values of quantities using word embeddings.  It includes the construction of a data set and some experiments with regression models.  The general direction of this work is worthy of study, but the paper needs additional justification for its task, better discussion of recent related work, and more development of its regression models.

The work starts by describing the construction of interesting crowdsourced data sets that include people’s estimates of typical quantities, what they would consider to be low or high values for a given object in given units e.g. the temperature of a hot spring or the height of a giraffe.  Overall, the data sets are interesting but are not especially large (2300 total [low, high] pairs of 230 different quantities).  Further, the particular task formulation here needs more justification.  I think most of us would agree that common sense is a critical AI challenge, and that the question of whether embeddings reflect typical quantities is important.  But, in a paper where the data set is considered a primary contribution, I would expect more justification for exactly how this task is formulated, and which objects and units were selected.  As one example, why ask about “large” and “small” values rather than something with more precise semantics (like the 10th and 90th percentile, for example)?  I also felt that the introduction could be improved to provide more convincing motivation.  E.g., the first paragraph only says that humans apply different adjectives (like “hefty” and “cheap”) to different things depending on their numerical attributes (weight, cost), but does not argue why teaching AI systems to use those adjectives is a priority.

Regarding related work, the paper is missing a discussion of several relevant papers that use embeddings to obtain relative comparisons or estimates of commonsense properties of objects, including:

Forbes, Maxwell, and Yejin Choi. "Verb physics: Relative physical knowledge of actions and objects." ACL 2017

Yang, Yiben, et al. "Extracting commonsense properties from embeddings with limited human guidance." ACL 2018

Elazar, Yanai, et al. "How Large Are Lions? Inducing Distributions over Quantitative Attributes." ACL 2019

The paper then presents the performance of some regression models.  These models are standard existing techniques, and given the relatively low performance I would have liked more development of the models and more analysis of the performance.  For a conference like ICLR I would expect to see a more thorough exploration and analysis of possible models for the task.  Looking at more powerful neural regressors (perhaps using contextual embeddings rather than just fixed word embeddings) might be one option.  Offering an explanation for why ARD seems to work better than the other approaches would be helpful.

Minor: In Table 3, the way that small and large are interleaved makes it hard to compare systems, I think presenting all the small results together, and large results together may help.  In Figure 2, it would be helpful to see the histogram for size-large within the same plots here, so we could see how far apart they are.

“Because Skip-gram has to handle more words to predict words, we assume Skip-gram will obtain more information about numerical values.”
-- I didn’t understand what you meant about skip-gram having to “handle more words to predict words.”  Also, I did not understand how this entails that skip-gram would obtain more info about numerical values.

**Experience Assessment:**

I have published one or two papers in this area.

**Review Assessment: Checking Correctness Of Derivations And Theory:**

N/A

**Review Assessment: Checking Correctness Of Experiments:**

I assessed the sensibility of the experiments.

**Review Assessment: Thoroughness In Paper Reading:**

I read the paper thoroughly.

---

### Official Review · AnonReviewer3 · 2019-10-25
**Official Blind Review #3**

**Rating:** 3

**Review:**

This paper describes the collection and analysis of a numerical common-sense dataset. The paper also states that a novel regression method is presented for quantifying numerical common sense.

Strengths:
- the numerical common sense task and its related challenges are presented and motivated clearly
- the data/annotations and their analysis are presented clearly. The experiment can be replicated reasonably (provided that the data is released, as the authors state)

Weaknesses:
- the dataset is too small (230 pairs of units and values). Two observations follow from this: (1) such small-scale data is of limited use to state of the art (deep learning) data-hungry approaches; (2) the relatively low cost of crowdsourcing typically results in much bigger datasets.
- some decisions or definitions seem a bit ad hoc and are not convincing. For instance: in Table 1, why is the apple measured in centimetres but the coffee cup measured in meters? Another example from section 1: why are temperature and weight described as nonphysical scales, but money described as a subjective scale? Another example of a definition: "similar words share similar units and thus concrete words contains numerical information". Statements like these are problematic because one could easily think of counter examples where this is not the case.
- the experimental findings are not based on state on the art methods applied in a setup where the robustness and transferability of the methods on other data and domains has been analysed and discussed.

Overall this work is interesting as a starting point. My advice is to consider scaling up both the data and the methods, and to accompany this by a more comprehensive analysis of limitations.

**Experience Assessment:**

I have read many papers in this area.

**Review Assessment: Checking Correctness Of Derivations And Theory:**

I assessed the sensibility of the derivations and theory.

**Review Assessment: Checking Correctness Of Experiments:**

I assessed the sensibility of the experiments.

**Review Assessment: Thoroughness In Paper Reading:**

I read the paper at least twice and used my best judgement in assessing the paper.

---

### Official Review · AnonReviewer2 · 2019-11-01
**Official Blind Review #2**

**Rating:** 1

**Review:**

This paper attempts to study if learned word embeddings for common objects contain information about "numerical common sense". The hypothesis is that certain numerical information may co-occur with the words for certain objects/measurement units within their context windows. To verify this hypotheses, the authors have created a dataset through a crowd-sourcing service which represents "numerical common sense". Using this dataset, the authors examine the predict abilities of regressors trained on learned word embeddings and the aforementioned crowd-sourced dataset. The hypothesis is that if the regressors demonstrate good accuracy, then the word embeddings contained information relevant to "numerical common sense". To the best of my knowledge, this is the first paper that attempts to analyze learned word embeddings in the context of numerical common sense.

This paper should be rejected because (1) the NCS datasets are too small to represent "numerical common sense" (2) the NCS datasets contain faulty data points and (3) the results from the experiments conducted are not sufficient to accept or refute the hypotheses.

Main argument

The first question that we must ask is - are the NCS-50x1 and NCS-60x3 datasets reliable for experiments on "numerical common sense". No, because of two flaws:

(1) The number of samples in the dataset is too small to represent "numerical common sense". Consider the histogram for object "dog" in Figure 2. If the largest data point in this plot was absent, the average of the distribution would be smaller by several orders of magnitude. Perhaps there are other objects in the dataset which are missing samples from the tail end of the distribution that could have large effects on the mean of the collected dataset.

(2) Some data points in the dataset don't make sense to me. For example, Fig 2 represents the "small" dataset, yet I see samples like 400m long dogs, 40m long cats, 150m long monitors and 20m long mice?

Also, it is not clear how the confidence scores of the participants were taken into account when training the regressors or if they were used at all.

If the NCS dataset does not represent "numerical common sense", it invalidates all experimental results from the paper.

My second issue with the paper is that it is not possible to conclude if the experimental results support or refute the hypothesis (ignoring the issue with the dataset):

1. In tables 2 and 3, the correlation coefficients were quite low and and the MAEs were pretty large. In Table 3, rows 1 and 2, even though the correlation is 0.57 and 0.48, the MAE is 100 million yen and 7.5 million yen respectively which is quite large. To me this suggests that just because the correlation is larger we cannot conclude mean that the model is performing well.

2. It is unclear why the correlation coefficient was chosen to decide that ARD is the superior model in experiment 1. The MAE for random forests with concatenated feature vectors was an order of magnitude smaller than that of the ARD model.

3. Why are the correlation coefficients missing for the unit-only experiment in Table 2? The LS model shows very good MAE relative to the other models and perhaps the correlation should have been measured for that as well? In fact, if the correlation coefficients for this case is comparable to the case with concatenated features, it would mean that the word embedding for the object is not helping at all! Moreover, I find it surprising that the LS model with concatenated features performs worse than the unit-only features. We cannot conclude if paper's interpretation about the results is correct unless this missing information is provided.

4. It is hard to judge what a correlation coefficient of 0.57 means. Why didn't you provide a scatter plot of the predictions vs targets as well? It often happens that even noisy plots demonstrate good correlations.

5. The paper should have additional ablation studies - for example, what would happen in the concatenated feature vector experiment if you trained the regressors using randomly initialized word embeddings instead of the trained word embeddings? Do you get the same performance as learned word embeddings?

**Experience Assessment:**

I do not know much about this area.

**Review Assessment: Checking Correctness Of Derivations And Theory:**

N/A

**Review Assessment: Checking Correctness Of Experiments:**

I assessed the sensibility of the experiments.

**Review Assessment: Thoroughness In Paper Reading:**

I read the paper at least twice and used my best judgement in assessing the paper.

---

### Decision · Program_Chairs · 2019-12-19

**Decision:**

Reject

**Comment:**

The authors tackle an interesting and important problem, developing numerical common-sense. They use a crowdsourcing service to collect a dataset and use regression from word embeddings to numerical common sense.

Reviewers were concerned with the size and quality of the dataset, the quality of the prediction methods used, and the analysis of the experimental results.

Given the many concerns, I recommend rejecting the paper, but I encourage the authors to revise the paper to address the concerns and resubmit to another venue.